

# Coping with shorter days: do phenology shifts constrain aphid fitness?

Jens Joschinski[1], Thomas Hovestadt[1,2] and Jochen Krauss[1]

[1] Department of Animal Ecology and Tropical Biology, Biocentre, University of Würzburg, Würzburg, Germany
[2] Department of Biology (TEREC), Ghent University, Ghent, Belgium

## ABSTRACT

Climate change can alter the phenology of organisms. It may thus lead seasonal organisms to face different day lengths than in the past, and the fitness consequences of these changes are as yet unclear. To study such effects, we used the pea aphid *Acyrthosiphon pisum* as a model organism, as it has obligately asexual clones which can be used to study day length effects without eliciting a seasonal response. We recorded life-history traits under short and long days, both with two realistic temperature cycles with means differing by 2 °C. In addition, we measured the population growth of aphids on their host plant *Pisum sativum*. We show that short days reduce fecundity and the length of the reproductive period of aphids. Nevertheless, this does not translate into differences at the population level because the observed fitness costs only become apparent late in the individual's life. As expected, warm temperature shortens the development time by 0.7 days/°C, leading to faster generation times. We found no interaction of temperature and day length. We conclude that day length changes cause only relatively mild costs, which may not decelerate the increase in pest status due to climate change.

Corresponding author
Jens Joschinski,
Jens.Joschinski@uni-wuerzburg.de

## INTRODUCTION

Nearly all organisms need to cope with environmental heterogeneity and fluctuation; showing a plastic response in the face of such heterogeneity can be beneficial. For example, several species from the *Daphnia* complex (Cladocera) can grow a 'crown of thorns' in response to predator pressure (*Petrusek et al., 2009*), and *Daphnia magna* allocates variable amounts of energy to size and shape as adaptive induced response to predator presence (*Rabus & Laforsch, 2011*). Similarly, many plants increase their investment into defence when attacked by herbivores (e.g., *Agrawal, 2011*). These examples demonstrate how phenotypic plasticity can affect fitness. One of the most important fitness traits is phenology (*Chuine, 2010*; *Helm et al., 2013*), i.e., the timing of life cycle events. Plasticity in phenology can profoundly change the ecology of a species, as it can alter the timing of critical life-history events and synchrony with other trophic levels (*Visser et al., 1998*; *Visser & Holleman, 2001*). Thus, phenological plasticity is an important component of the ecology and evolution of species.

Phenotypic plasticity can not only be adaptive in temporally fluctuating environments, but also prevent extinction in environments under directional change (*Chevin et al., 2013*). The current rate of environmental change is likely unprecedented in the last 1,400 years (*IPCC WG I, 2013*), as the global surface temperature rises by 0.2 °C per decade (*Hansen et al., 2006*). Climate change modifies the onset and duration of seasons (*Räisänen & Eklund, 2012*), and many species have already responded by shifting their phenology in the according direction (*Rosenzweig et al., 2007*). By adjusting phenology *via* plastic responses, organisms can possibly mitigate the extinction risk imposed by climate change (*Charmantier et al., 2008*; *Vedder, Bouwhuis & Sheldon, 2013*), and even profit from it (*Bell et al., 2015*).

However, the evolution of phenotypic plasticity may be constrained by costs and limits (*DeWitt, Sih & Wilson, 1998*). For example, plasticity can be limited by tightly interacting species, which may not shift their timing in synchrony. Among the best studied examples are great tit populations which have lost synchrony with their caterpillar prey (*Visser et al., 1998*), and winter moths which are no longer synchronous with their host (*Visser & Holleman, 2001*). We hypothesize that another limit of plasticity is posed by the reduction in day length (photoperiod) associated with a shift in phenology: First, activities of a diurnal species, e.g., foraging, can be constrained by shorter days, if individuals live in a later time of the year. Secondly, photoperiod is the most common cue to predict seasonal change (*Saunders, 2013*). Photoperiodism is commonly assumed to be based on the circadian clock (*Bünning, 1936*; *Saunders, 2013*), a molecular clockwork which governs rhythmicity (*Peschel & Helfrich-Förster, 2011*). Thus, we hypothesize that altered day length conditions interfere with the (yet unresolved) interplay of seasonal and circadian rhythmicity and hence affect phenotypic plasticity.

The effect of warming temperature on fitness is relatively well established. Within physiological limits warmer temperature generally speeds up metabolic rates (*Gillooly et al., 2001*). Less researched, and potentially important in a changing climate, are interactions of day length and temperature. We propose that warmer temperature results in faster growth during the organism's active period, but higher energy expenditure during resting time. Hence, the effect of temperature should depend on day length. Also, temperature might enhance the interference with circadian timing, as the clockwork is not fully compensated for temperature changes (*Saunders, 2014*). Thus, short day conditions may decrease insect fitness, whereas warm temperature should enhance growth rates, and warming might enhance the fitness costs of short days.

Aphids like *Acyrthosiphon pisum* (HARRIS) are well suited to study constraints of short days. During summer *A. pisum* reproduces clonally, establishing exponentially growing populations. However, live-born nymphs have little chance to survive sub-zero temperatures (*Simon, Rispe & Sunnucks, 2002*). Therefore, in many clones aphids give birth to a single generation of sexual morphs in autumn, which produce cold-resistant eggs to overwinter. In warmer climates this response to photoperiod is frequently lost, so asexual aphid morphs are active throughout the year (*Simon, Rispe & Sunnucks, 2002*). These differences in phenology within one species allow studying day length effects in a seasonal insect without actually inducing a photoperiodic response.

Specifically, we hypothesize that:

(1) Shorter day length constrains aphid performance and reduces population growth.
(2) Warm temperature causes quicker generation cycles and faster population growth.
(3) Temperature and day length interact, so that the positive effects of an increase in ambient temperature decline with shorter day length.

We therefore expect fitness costs under short-day conditions compared with long-day conditions, and possibly the lowest fitness under short days combined with warm conditions.

## MATERIALS AND METHODS

To test for constraints of phenotypic plasticity, we carried out experiments with an asexual clone of the aphid *A. pisum* in four climate chambers at the individual as well as at the population level. We measured population growth on whole plants of *Pisum sativum* (L.), and life history data of individuals raised on cut leaves of *P. sativum*.

### Day length and temperature settings

We used four identical climate chambers (Sanyo/Panasonic MLR-H series), in which we applied two realistic temperature settings with sinusoid day/night cycles, ranging from 12 to 23 °C (±1 °C) and from 14 to 25 °C (±1 °C), and two day length regimes with day length of 12:12 LD and 16:8 LD (Fig. 1), using 40 W fluorescent lamps. The temperature differed between the light treatments at dawn and dusk, but this difference in light sums is only 1.2%. Treatments were exchanged weekly, because the maximum light intensities varied between chambers from 13,000–21,000 lux. Because development and reproductive period lasted four weeks, all treatments received the same light sum (lux * h) over this period. The lower temperature settings in the experiment approximately reflect naturally occurring temperatures in Würzburg, southern Germany, during summer solstice (12–22 °C) and during beginning of September (11–22 °C; data from Deutscher Wetterdienst, http://www.dwd.de/). The higher temperature settings simulate climate change with moderately increased mean temperature of 2 °C, which ranges between the SRES B1 and B2 marker scenario projections for 2099 (*IPCC WG III, 2000*). We are aware that this is a conservative estimate; nevertheless, we used this low difference of means so that we did not confound the results by exceeding the physiological optimum of the pea aphid.

### Study organisms

Due to its fast population growth and its properties as virus vector, *Acyrthosiphon pisum* (HARRIS, Aphididae) is a pest in agriculture, which is distributed throughout northern Europe, North America and New Zealand (*Blackman & Eastop, 2000*), *Acyrthosiphon pisum* feeds on legume crops such as pea (*Pisum sativum* L.) and bean (*Vicia faba* L.), and does not switch hosts in autumn. The aphid clone L1_22, an asexual green alfalfa biotype, was kindly provided by Grit Kunert (MPI Jena). The known asexuality of the clone has been confirmed by providing an 8:16 LD rhythm at 10 °C for four generations.

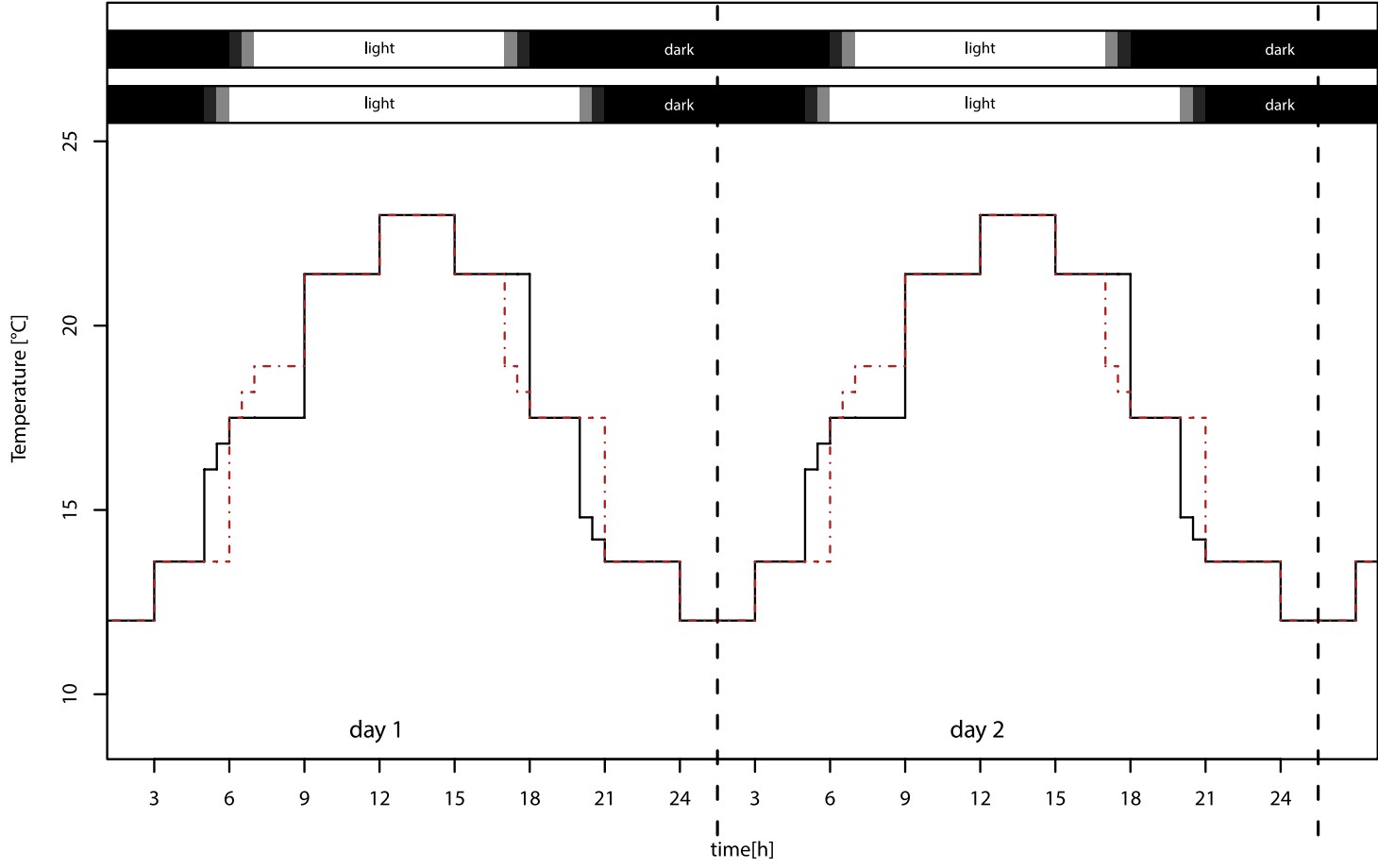

**Figure 1  Temperature settings of the climate chambers.** Warm temperature settings for long day (solid black lines, lower bar) and short day (dashed red lines, upper bar) treatments. Mean temperatures of long and short day conditions do not differ. The temperature in the two low temperature treatments was overall 2 °C lower (not shown).

*Pisum sativum* L. is a suitable host plant for *A. pisum*, and agricultural plants are frequently attacked by aphids (*Blackman & Eastop, 2000*). We used the breed 'Kleine Rheinländerin' (Bingenheimer Saatgut, Echzell, Germany), which grows to 40 cm, for all experiments.

## Performance of individual aphids

To detect day length and temperature effects on the individual performance of aphids, we placed 20 adult apterous, asexual aphids per climate chamber ($20 \times 4 = 80$) singly in plastic tubes ($8 \times 3.5$ cm), and used their first born nymphs (termed first generation) as new focal individuals for further measurement. These first-generation nymphs were fed every second day with one cut leaf each, and we recorded development time, length of reproductive period, post-reproductive period and life span. We used cut leaves like *Meister et al. (2006)* to exclude differences in food quality, as a living host plant can be expected to fix more carbon under long day conditions. We counted and discarded newly born nymphs daily (thus measuring daily fecundity and lifetime reproductive output of each

focal animal). In order to test for maternal effects in a second generation and to confirm the loss of sexuality in the clone, we retained one early-born nymph per focal aphid after 11–13 days. Additionally, we retained one late-born nymph after 29–31 days because we expected the maternal effects to intensify as the adult ages. We raised all aphids of the second generation under the same conditions (16:8 LD, 22 °C and 60% humidity), so that maternal effects could be distinguished from direct effects of day length and temperature. These second generation aphids were fed with fresh plant material every second day, and life history parameters were also recorded every second day.

To supply the aphid individuals with food we grew 60 pea plants ('Kleine Rheinländerin') per week with two plants per pot (11 × 11 cm, filled with Einheitserde® classic soil; Einheitserdewerk Hameln GmbH, Sinntal, Germany) over six weeks at 22 °C, 16:8 LD and 60% humidity, so that 2–3 week old plant material (approximate BBCH growth stage 14–15) was available over the whole course of the experiment. Pea plants grow pinnate compound leaves with morphologically different stipules. We fed four leaflets from the same leaf compound (the youngest which had completely unfolded leaflets), but excluded the basal stipulate leaves. If there was not enough plant material available, we fed the aphids with plant material of two leaf compounds of similar age. The four leaflets were randomly distributed over the four treatments to ensure that all treatments received the same plant quality. We used the same plant no more than twice in order to avoid induction of defense. The plants were always raised at 22 °C and in a 16:8 LD cycle.

Altogether 80 individuals of the aphid *A. pisum* were used in the experiment. Nine aphids died before reaching reproductive age, and six individuals (7.5%; five under cold, short day and one under warm, short day treatment) developed into alate (winged) virginoparous morphs. The 15 deceased or winged individuals were excluded from further analysis. A further ten aphid individuals were accidentally killed as adults, which reduced the number of replicates to 55 aphids for the traits fecundity, reproductive period, post-reproductive period and life span.

## Population experiment

To detect the effects of day length and temperature on population demography, we sowed 60 pea plants into 11 × 11 cm square pots filled with a peat-based substrate (Einheitserde® classic; Einheitserdewerk Hameln GmbH, Sinntal, Germany). The plants were watered from above during the first week and from below (using felt mats) thereafter in four trays with 15 plants each. We kept all plants in a walk-in climate chamber with 22 °C at 16:8 LD and 60% humidity and watered five times per week. After 18 days, we fixed each plant with raffia fibres to 50 cm wood sticks. After 25 days, 12 plants from each tray were evenly distributed over the four climate regimes (48 plants in total), and the position within each chamber fully randomized. Following one week of acclimation, we established aphid populations by placing 10 individuals of adult apterous (wingless) asexual morphs on each individually bagged plant, using micro-perforated plastic bags (255 × 700 mm, 0.5 mm perforations, Baumann Saatzuchtbedarf, www.baumann-saatzuchtbedarf.de).

To accommodate for climate chamber differences, we exchanged treatments between chambers weekly. We estimated population size weekly by counts of alate (winged) and apterous adults and nymphs (judged by the visibility of the cauda and size differences) over the course of four weeks on the living plants (BBCH growth stages approximately 16–19). To control the effect of heat stress on the plants, we distributed 24 aphid-free, 23 days old plants over the four chambers to observe plant responses to the artificial climate over four weeks.

## Statistics

We used R version 2.15.2 (*R Core Team, 2012*) for all analyses. On the individual level, 65 out of 80 aphids were used to assess development time, and 55 for the remaining variables (length of reproductive period, length of post-reproductive period, life span and fecundity). We tested effects of day length, temperature and their interactions as main factors in two-way ANOVAs on all of those parameters except fecundity. The latter we used to construct a Leslie Matrix to yield the theoretical population rate of increase $r_t$ and the reproductive values of each age cohort (*Leslie, 1945*). We used a Leslie matrix because averaged daily fecundity (as for example used by *Meister et al., 2006*) does not account for skews in the fecundity curve, which cause shorter generation times and alter growth rate projections. In particular, late-born offspring add very little to population growth compared to early-born offspring, and the true fitness costs may be over- or underestimated. We used the estimates of $r_t$ in a two-way ANOVA to also test for effects of day length and temperature. At the population level, we calculated the weekly population growth rates $r_1$, $r_2$ and $r_3$ on 48 plants, as ($N_x/N_{x-1}$), using the aphid number $N$ at week $x$, and the daily growth as $r_x^{(1/7)}$. We compared the rates of increase, i.e., log(growth rates), in two-way ANOVAs as before. Because a temperature gradient existed within the climate chambers, the position within chambers had a significant effect for nymphal development and $r_t$. However, as the position effect was in the same direction as the effect of temperature and did not qualitatively change the results, we omitted it from analysis.

## RESULTS

### Life history traits of individual aphids

In our experiment, aphids developed on average within $10.7 \pm 0.2$ days and warm temperature shortened the development time significantly (Fig. 2, Tables 1 and 2). The length of the reproductive period (Fig. 2 and Table 1) and the fecundity of aphids (Fig. 3 and Table 1) depended solely on day length. Aphids raised under short-day conditions reproduced about 3 days (14%) less, and produced 22% fewer nymphs (Table 2). The post-reproductive period ranged from $5.0 \pm 0.6$ (warm, long) to $9.8 \pm 1.3$ (cold, short) days, and was elongated by a reduction of day length and of temperature (Fig. 2). Overall, warm temperature shortened the total life span, i.e., the sum of development time, reproductive and post-reproductive period. Even though the food quality was sufficient for full development (including the post-reproductive period) of all focal aphids in the first generation, the second generation suffered high mortality rates (34%) and reduced

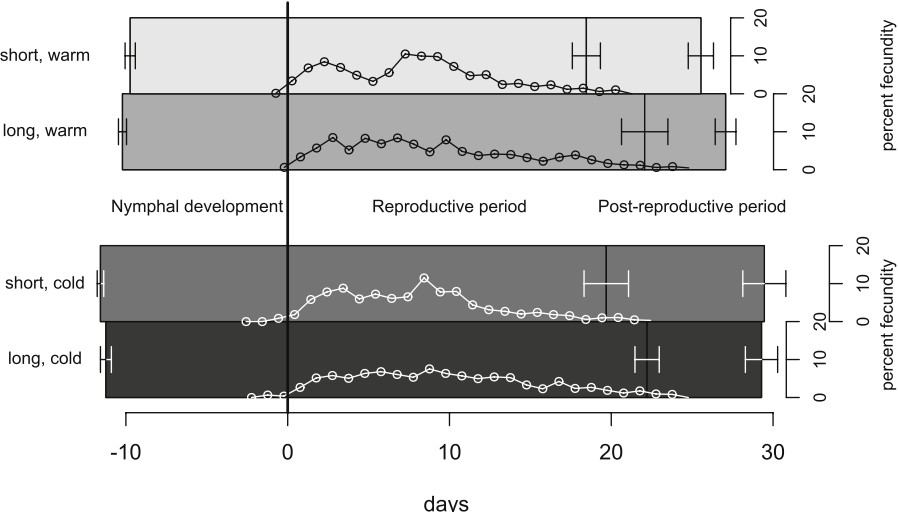

**Figure 2 Life-history traits of individuals reared under different climate conditions.** The bars are aligned at the mean onset of reproduction (i.e., not left-aligned) to better distinguish temperature effects (on development) from day length effects (on reproduction). Bars indicate S.E. Lines with open circles indicate the timing of nymph production (expressed as daily contributions to total fecundity in %). These curves form also the basis for the Leslie calculations (Tables 1 and 2). Statistics see Table 1.

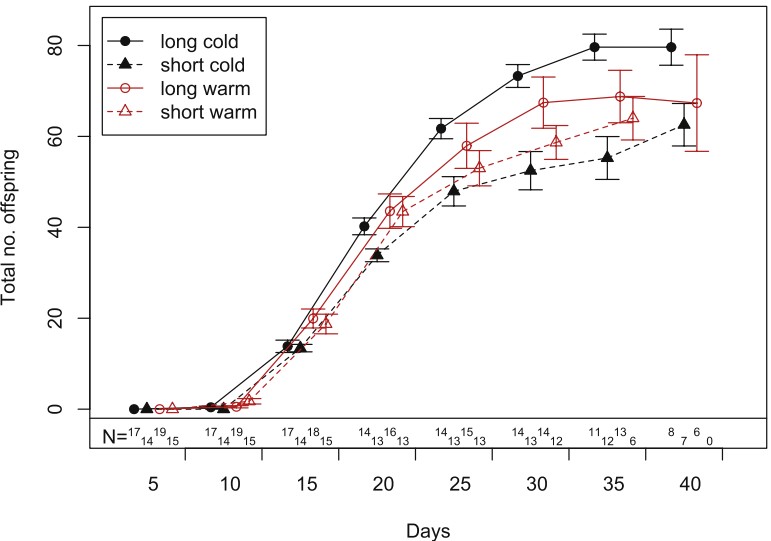

**Figure 3 Cumulative fecundity as function of age of individuals reared under four climate conditions.** Bars indicate S.E. Statistics see Table 1. Sample size (*N*) declines over time, because the aphid mortality increases with age (c.f. Fig. 2).

offspring numbers (to 0–30%). Seventy-three out of seventy-five surviving adults of the second generation reproduced and no males were observed; we therefore confirm that the focal aphids did not switch from asexual to sexual offspring. The theoretical population rates of increase $r_t$ (based on Leslie matrices) differed significantly between temperature regimes, but were independent of day length (Fig. 4A). The reproductive values of the last three days of reproduction were on average 1.56, which is 9.7% of the maximum

**Table 1** ANOVA tables testing for day length and temperature effects on aphid life history traits.

| Response variable | Factor | F | df | p (<F) |
|---|---|---|---|---|
| **Development time** | Temperature | 23.62 | 3,64 | **<0.001** |
| | Day length | 0.10 | 3,64 | 0.759 |
| | Temp × day length | 2.01 | 3,64 | 0.162 |
| **Reproductive period** | Temperature | 0.27 | 3,54 | 0.603 |
| | Day length | 6.98 | 3,54 | **0.011** |
| | Temp × day length | 0.22 | 3,54 | 0.643 |
| **Post-reproductive period** | Temperature | 6.36 | 3,54 | **0.015** |
| | Day length | 6.22 | 3,54 | **0.016** |
| | Temp × day length | 0.11 | 3,54 | 0.747 |
| **Life span** | Temperature | 9.24 | 3,54 | **0.004** |
| | Day length | 0.33 | 3,54 | 0.567 |
| | Temp × day length | 1.22 | 3,54 | 0.274 |
| **Total fecundity** | Temperature | 1.33 | 3,54 | 0.253 |
| | Day length | 12.84 | 3,54 | **<0.001** |
| | Temp × day length | 2.70 | 3,54 | 0.107 |
| **$R_t$ (rate of increase derived from life- history traits)** | Temperature | 6.90 | 3,54 | **0.011** |
| | Day length | 0.08 | 3,54 | 0.773 |
| | Temp × day length | 2.95 | 3,54 | 0.092 |
| **Population rate of increase** | Temperature | 4.92 | 3,41 | **0.032** |
| | Day length | 0.04 | 3,41 | 0.836 |
| | Temp × day length | 0.54 | 3,41 | 0.465 |

**Notes.**
Significant effects are shown in bold.

**Table 2** Effect sizes of the four day length/temperature treatments on aphid life history traits.

| Response variable | | Short day | Long day |
|---|---|---|---|
| **Development time (days)** | Low temp | 11.6 (±0.2) | 11.2 (±0.3) |
| | High temp | 9.7 (±0.3) | 10.2 (±0.3) |
| **Reproductive period (days)** | Low temp | 19.7 (±1.4) | 22.2 (±0.8) |
| | High temp | 18.5 (±0.9) | 22.1 (±1.4) |
| **Postreproductive period (days)** | Low temp | 9.8 (±1.3) | 7.1 (±1.0) |
| | High temp | 7.1 (±0.8) | 5.0 (±0.6) |
| **Life span (days)** | Low temp | 41.1 (±1.2) | 40.3 (±1.4) |
| | High temp | 35.0 (±1.1) | 37.4 (±1.8) |
| **Total fecundity (nymphs)** | Low temp | 54.2 (±4.7) | 77.7 (±2.9) |
| | High temp | 56.5 (±4.1) | 65.3 (±5.5) |
| **$R_t$ (rate of increase derived from life-history traits)** | Low temp | 0.23 (±0.003) | 0.24 (±0.004) |
| | High temp | 0.26 (±0.009) | 0.25 (±0.010) |
| **Population rate of increase** | Low temp | 0.24 (±0.010) | 0.25 (±0.017) |
| | High temp | 0.31 (±0.012) | 0.29 (±0.039) |

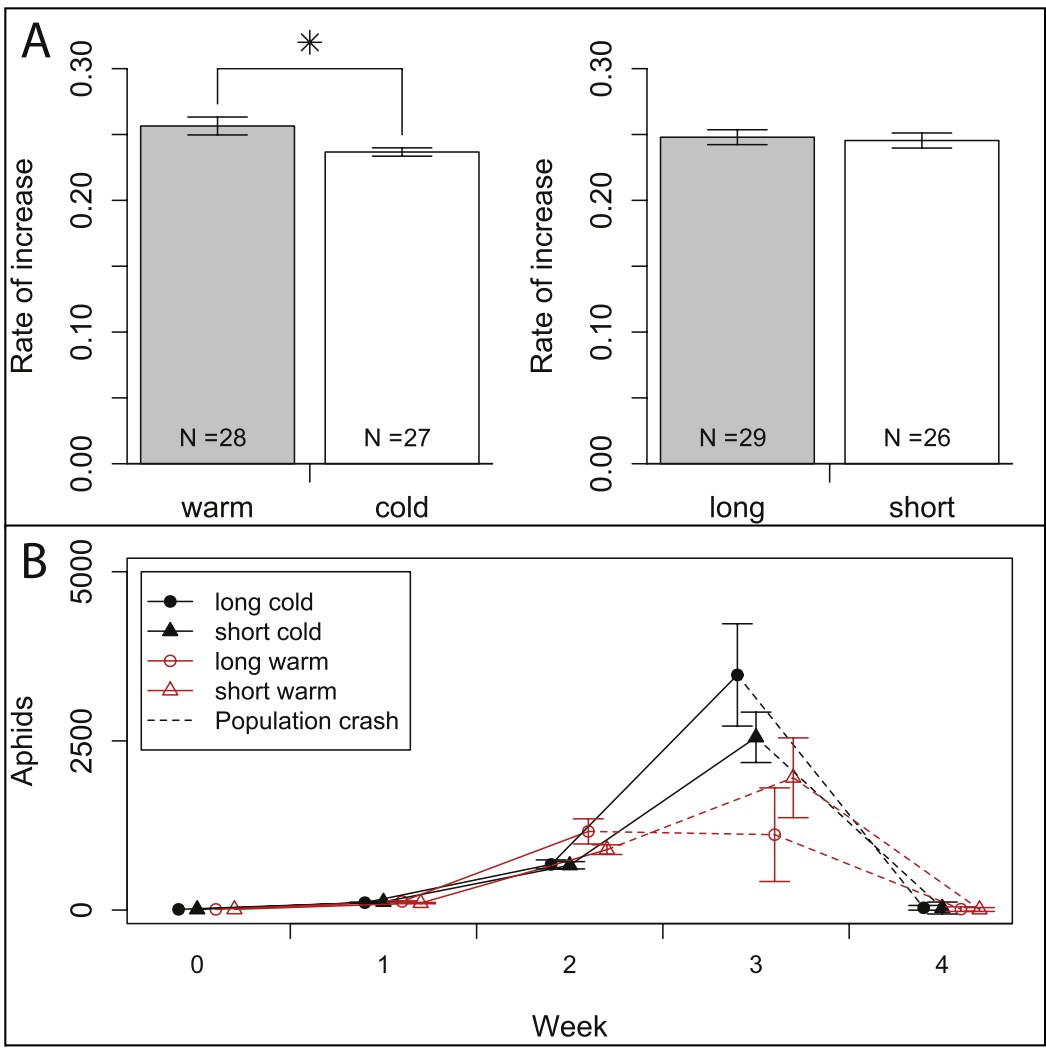

**Figure 4 Growth rates of aphids under warm and cold conditions at 16:8 h and 12:12 h day length.** (A) Comparison of population rates of increase under warm vs. cold and under long day vs. short day conditions. Data is based on Leslie-matrices derived from individual life histories. Bars indicate S.E. (B) Population growth of aphids reared on whole plants. Dashed lines indicate the time when half of the plants died, presumably from increasing pest load. Statistics see Table 1.

reproductive value (16.99). The average growth rate was below the growth rate of the population experiment (see 'Fitness costs on the population level'), possibly because the cut leaves do not provide enough phloem pressure.

## Fitness costs on the population level

In the population experiment, about 10% of the observed aphids were adults, and 0% (in the first two weeks) to 13% (in the third week) of the adults were winged. Adult/nymph ratios and winged/wingless ratios never varied significantly among treatments (all $p > 0.1$), so differences in wing induction patterns are unlikely to have affected our results. Aphid density (sum of nymphs, winged and wingless aphids) increased exponentially over the first two weeks, with a weekly growth of about one order of magnitude (Fig. 4B). After two

weeks, aphid densities were higher in the warm treatment (1,027 ± 101 aphids) than in the cold treatment (668 ± 42 aphids), but not significantly affected by day length (Table 1). In week three, the exponential growth ceased, and during weeks three and four most plants died and aphid densities declined, especially in the warmer treatments. Control plants without aphids did not show any signs of heat stress and were healthy throughout the experiment.

## DISCUSSION

Plasticity in phenology likely helps to make use of novel climate conditions and to extend the asexual season, which may increase the pest status of aphids (*Bell et al., 2015*). However, the novel day length conditions under which the animals live may be non-optimal to the organism, and thus reduce the advantage of plasticity. Our results show that a 2 °C increase in temperature accelerates development and increases the population growth in an asexual aphid clone, but does not alter the individual reproductive period or fecundity. In contrast to increased temperature, a shorter day length reduced the length of the reproductive period by 14% and fecundity by 22%, but did not significantly affect development time or life span.

### Day length

In our experiment, day length alters fecundity and length of the reproductive period, and aphids suffer under short-day environments from reduced reproduction.

Even though variation in phenological traits is commonly regarded as phenotypic plasticity (*Charmantier et al., 2008*; *Vitasse et al., 2010*; *Vedder, Bouwhuis & Sheldon, 2013*), the microevolutionary costs and limits of plasticity (sensu *DeWitt, Sih & Wilson, 1998*) in phenology have to our knowledge never been measured. Phenotypic plasticity in phenology often relies on day length (photoperiod) as cue, and our study is the first that demonstrates fitness costs linked to short days in insects. On living plants, aphids exhibit circadian rhythmicity and seem to be day-active (*Eisenbach & Mittler, 1980*; *Hodgson & Lane, 1981*; *Cortes, Ortiz-Rivas & Martinez-Torres, 2010*), which offers—in agreement with the hypothesis outlined in the introduction—a tentative explanation for the observed fitness loss under short days. Further studies will need to verify the diurnality independent of host plants, and to measure phloem consumption under long and short days.

Photoperiod may also have a less direct effect on fitness, as its measurement may be based on the circadian clock (*Bünning, 1936*), an endogenous time-keeping mechanism which relies on two cyclically expressed protein complexes, PERIOD/TIMELESS and CLOCK/CYCLE (*Peschel & Helfrich-Förster, 2011*). Interference among seasonal rhythm and circadian clock seems reasonable, though this hypothesis is still under debate (*Danks, 2005*; *Kostal, 2011*). Hence, shortening day length may not only affect the time available, but also its correct measurement. So far, relatively little is known about the circadian rhythm of aphids, but with the recent identification of the clock genes in aphids (*Cortes, Ortiz-Rivas & Martinez-Torres, 2010*), further progress can be expected.

On the population level we did not detect effects of day length on fitness. Our calculation based on Leslie matrices indicates that short day length does not significantly

dampen population growth, because the additional offspring produced under long days are born rather late in the adults' life (c.f. Fig. 2); thus, only life stages with little reproductive value are affected. Consequently, substantial costs of shortened day length are not observed in our population experiment. We thus conclude that the observed reduced reproduction does not impede population growth.

## Temperature

As expected, we found that warmer temperature shortens the life cycle of aphids. Because the quicker life cycle leads to faster population growth both in our Leslie calculations and on real plants, climate change with increased mean temperatures should increase the pest potential of aphids (*Bell et al., 2015*). Presumably warmer temperature acts on metabolic rates, as is well established for insects (*Gillooly et al., 2001*). Temperature did, however, not change fecundity or the length of the reproductive period over the measured range, and thus warm temperature *per se* does not affect an individual's condition. This contradicts studies of temperature on the condition of *A. pisum* by *Campbell & Mackauer (1977)* and *Kaakeh & Dutcher (1993)*, but supports the results of *Kilian & Nielson (1971)*. On a different aphid species, *Rispe, Simon & Pierre (1996)* also detected no general effect of temperature on fecundity, but large variation among clones. Clonal variation also explains differences between the cited experiments.

Because variability in temperature will likely increase due to climate change (*Solomon et al., 2007*), we included diurnal cycles in our design. Due to the nonlinear shape of the growth rate curve, variability should increase the growth rate as long as it is below the optimum (*Estay, Lima & Bozinovic, 2013*). Several studies on other clones indicate that the physiological optimum of *A. pisum* lies beyond 20 °C, and decreases only at temperatures higher than 25 °C–30 °C (*Kenten, 1955*; *Kilian & Nielson, 1971*; *Campbell & Mackauer, 1977*; *Kaakeh & Dutcher, 1993*; *Rispe, Simon & Pierre, 1996*). Our treatments lie with 17.5 and 19.5 °C below the reported optimum, so one would expect a larger effect of an increase in mean temperature on reproductive traits in our experiment compared to experiments applying constant temperatures. However, this hypothesis was not supported by our experiment. *Kilian & Nielson (1971)* and *Kaakeh & Dutcher (1993)* recorded with constant temperatures around similar means (15/20 °C) a shortening of development time by 0.7 and 1.2 days/°C, respectively. These values are largely in line with those in our experiment, where the onset of reproduction shifted by 0.7 days/°C. However, we found some effect of temperature variability on longevity because in our study, in contrast to *Kilian & Nielson (1971)*, life span decreased under long days by 1.4 days/°C. It is possible that, our clone is adapted to colder temperature, so that the maximum temperatures of 25 °C stressed the aphids and caused a hazard. Therefore, higher temperature variability may decrease, not increase, aphid performance.

Contrary to our hypothesis that temperature has opposing effects at day and night, we found no interaction of day length and temperature. We hence conclude that day- and night time temperatures have similar effects on aphid fitness and impose physiological constraints only by generally affecting the aphid metabolism.

## CONCLUSION

We show that a shorter photoperiod reduces reproduction in obligately asexual aphids. Consequently, the aphids' potential benefits following from global change are reduced, as temperature increase may lead to novel day length-temperature correlations. If the fitness decline has its roots in physiological constraints, our results may be extrapolated to any day-active insect species. However, these side-effects of phenotypic plasticity were not detected at the population level because they affect only late fitness components in the individual's life. We further show that warm temperatures increase aphid growth by shortening development, but neither reduce individual reproduction, nor do they modulate the effect of short day length. Taken together, we conclude that novel light: temperature relations do not suppress the pest potential of aphids in a changing climate.

## ACKNOWLEDGEMENTS

We thank Grit Kunert, MPI Jena, for provision of the aphid clone, and we thank Christie Bahlai and two anonymous reviewers for useful comments on the manuscript.

### Funding

Funding was provided by the German Research Foundation (DFG), collaborative research center SFB 1047 "Insect timing," Projects C3 and C6. The funders had no role in study design, data collection and analysis, decision to publish, or preparation of the manuscript.

### Grant Disclosures

The following grant information was disclosed by the authors:
German Research Foundation (DFG).
Collaborative research center: SFB 1047.

### Competing Interests

The authors declare there are no competing interests.

### Author Contributions

- Jens Joschinski conceived and designed the experiments, performed the experiments, analyzed the data, wrote the paper, prepared figures and/or tables.
- Thomas Hovestadt analyzed the data, contributed reagents/materials/analysis tools, reviewed drafts of the paper.
- Jochen Krauss conceived and designed the experiments, contributed reagents/materials/analysis tools, reviewed drafts of the paper.

### Supplemental Information

Supplemental information for this article can be found online at http://dx.doi.org/10.7717/peerj.1103#supplemental-information.

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
