# Peer review of "Coping with shorter days: do phenology shifts constrain aphid fitness?"

_PeerJ, doi:10.7717/peerj.1103_

## Round 0.1 · original submission · Minor Revisions

AS you can see from the reviewers comments there are only a few minor issues that I am sure you will be able to deal with.

·

Basic reporting

Looks good

Experimental design

Looks good

Validity of the findings

Good, just one thought on interpretation- see general comments.

Additional comments

This study is useful, competently completed and interesting. I only have a few comments that I would like the authors to consider.

I notice there’s a few issues with sigfigs throughout the MS. Standard errors should be rounded to one significant digit, and means should be rounded to that same digit. - here’s a link with an explanation of that: http://anderson.cm.utexas.edu/AISD/SignificantDigitsAndStandardDeviation.pdf

A. pisum, like most aphids, is well known for phenotypic plasticity, but this is usually used to refer to traits that have some sort of irreversibility- ie alate vs apterous morphs. I feel that the concept of phenotypic plasticity is somewhat oversold in the manuscript because of this convention. I appreciate that, under the definition of phenotypic plasticity, any alteration to the biology of an organism in response to environmental conditions is an example, but it leaves me wondering, is what’s being measured here actually phenotypic plasticity, or is it physics? As a general rule, when you heat an insect up, it goes faster (within limits), because its metabolism is tied to temperature. If you take an individual from one temperature regime and put it in a cooler temperature regime, you’ll generally find that whatever trait you’re measuring for will happen slower. I don’t feel that’s necessarily a change in phenotype- If you measure things in metabolic time (ie degree days), you may find the organism is responding exactly the same way. What I think qualifies as phenotypic plasticity in this situation would be differences in slope of response to temperature- say, in response to different day lengths. I’m not sure exactly how the statistics were encoded, and it’s not ideal as there were only two temperature treatments but I think a better way to look for phenotypic plasticity is to compare either temperature response slopes in the two daylight regimes, or compare degree day accumulation by day length, etc.

Figure 4 A Broken y axes are used to show differences more effectively, but I find they’re misleading- see: http://faculty.atu.edu/mfinan/2043/section31.pdf
B- the X axis is very difficult to understand. I suggest simplifying. Also, I’m having trouble understanding why the lines joining the points after the population crash are dashed.

Reviewer 2 ·

Basic reporting

I think the paper is technically OK. The findings are not very novel - it is a standard experiment showing life-history traits of individuals reared under different climate conditions.

Experimental design

The experimental design is simple and I do not see any problems here.

Validity of the findings

As I said before, the data were correctly sampled and analyzed. The experiment was simple, and therefore I do not see any problems with this.

Reviewer 3 ·

Basic reporting

This is an interesting paper dealing with a well-studied aspect of aphid biology, the interaction between temperature and photoperiod, but placing it in the context of climate change and host plant synchrony. There are a few minor issues that need clarifying before publication. In particular I would urge the authors to read a bit more of the relevant literature.

Experimental design

Line 113 from what part of the plant were the leaves taken and at what growth stage were the plants? Aphid reproduction and development are greatly affected by host plant growth stage (e.g. Dixon, A.F.G. & Wellings, P.W. (1982) Seasonality and reproduction in aphids. International Journal of Invertebrate Reproduction, 5, 83-89. Maiteki, G.A. & Lamb, R.J. (1985) Growth stages of field peas sensitive to damage by the pea aphid, Acyrthosiphon pisum. Journal of Economic Entomology, 78, 1442-1448.)

Line 145 at what growth stage were the plants when placed in the array? Use BCCH stage so that other will be able to replicate your experiment (Leather, S.R. (2010) Precise knowledge of plant growth stages enhances applied and pure research. Annals of Applied Biology, 157, 159-161.).

Validity of the findings

No comments

Additional comments

Line 258 I would refer the authors to Kenten, J. (1955) the effect of photoperiod and temperature on reproduction in Acyrthosiphon pisum (Harris) and on the forms produced. Bulletin of Entomological Research, 46, 599-624, who effectively did the same experiments but under constant temperature conditions. They may also like to read Lamb, R.J. & Mackay, P.A. (1988) Effects of temperature on development rate and adult weight of Australian populations of Acyrthosiphon pisum (Harris) (Homoptera: Aphididae). Memoirs of the Entomological Society of Canada, 146, 49-55 and Lamb, R.J. & Mackay, P.A. (1997) Photoperiodism and life cycle plasticity of an aphid, Macrosiphum euphorbiae (Thomas), from central North America. Canadian Entomologist, 129, 1035-1048, who also did very similar work.

---

## Round 0.2 · accepted · Accept

Thank you for making the correction clear and for publishing in PeerJ, I look forward to handling your next manuscript.